# The Relationship between Postmenopausal Women’s Self-Esteem and Physical Activity Level—A Survey Study from Poland

**DOI:** 10.3390/ijerph19159558

**Published:** 2022-08-03

**Authors:** Magdalena Dąbrowska-Galas, Agnieszka Drosdzol-Cop

**Affiliations:** 1Department of Kinesitherapy and Special Methods, School of Health Sciences in Katowice, Medical University of Silesia, 40-752 Katowice, Poland; 2Department of Gynecology, Obstetrics and Gynecological Oncology, Medical University of Silesia, 40-752 Katowice, Poland; adrosdzol@sum.edu.pl

**Keywords:** postmenopause, IPAQ, SES

## Abstract

Introduction: Physical inactivity has become one of the major public health and economic concerns in Western societies. The consequences of physical inactivity are associated with many physical problems, however, the influence of physical activity (PA) on psychological health is unclear. The aims of our study were to assess self-esteem and physical activity levels in postmenopausal women and to examine the association between physical activity levels and self-esteem in this group. Material and methods: Survey research was conducted on postmenopausal women aged M = 58.81 ± 7.68 in women’s health clinics in Silesia, Poland. The total number of participants was 131, and 18 were excluded. A questionnaire with socio-demographic data and other international questionnaires were used: International Physical Activity Questionnaire, Rosenberg Self-Esteem Scale and Beck Depression inventory. Results: 78.76% of postmenopausal women were physically active. Mean value of MET-min/week was M = 1543.46 ± 1060.92. A total of 11.51% of women reported low self-esteem, with the mean total value of SES M = 31.79 ± 2.93. There was a lack of correlation between total IPAQ score and self-esteem (r = −0.241, *p* = 0.01). Conclusions: Postmenopausal women have mostly average self-esteem. They are generally active, and walking is the most common form of physical activity, however, a higher PA level does not influence self-esteem.

## 1. Introduction

Physical activity (PA) is defined as a behavior that involves human movement, resulting in physiological attributes including increased energy expenditure and improved physical fitness [1]. Adults are classified as inactive if they did not perform any session of light to vigorous leisure-time PA of at least 10 min per day [2]. Another definition says that individuals who walk fewer than approximately 3000 steps or expend fewer than 1.5 kcal/kg/day are considered to be inactive [3].

Physical inactivity has become one of the major public health and economic concerns in Western societies. In 2013, worldwide, the global economic cost of physical inactivity for the health care system was estimated at around USD 54 billion [4]. In 2017, the World Health Organization (WHO) set out a plan (Global Action Plan on Physical Activity) to promote physical activity and reduce physical inactivity by 10% by 2030. However, in 2018, it was reported that the goal will not be achieved [5,6]. Worldwide, 1 in 4 adults do not meet physical activity recommendations. In some countries, due to changing patterns of transportation, increased use of technology and urbanization, the level of inactive adult population increased to 70% [6]. The WHO recommends that adults achieve a minimum of 150 min of moderate physical activity per week. The consequences of physical inactivity, in particular the increased risk of many adverse health conditions such as obesity, morbidity and mortality, are the impetus for many public health campaigns [7]. Physical inactivity is responsible for approximately 10% of type 2 diabetes, coronary heart disease (CHD) and breast and colon cancers worldwide. The elimination of physical inactivity would remove approximately 10% of major CHDs, type 2 diabetes and breast cancer [8]. The increasing trend of a higher prevalence of obesity in women worldwide is shown in epidemiological data [9]. It is well-known that low estrogens levels, due to ovarian insufficiency, are associated with decreased energy expenditure, leading to long-term weight gain and obesity [10]. After menopause, the lean mass declines, the rate of fat tissues increases, and most women complain about weight gain [11]. During the menopausal transition, women gain an average of 2–3 kg, and a shift to an abdominal fat redistribution occurs [12]. It was estimated that postmenopausal women had a 4.88-fold higher risk of developing abdominal obesity compared with premenopausal [13]. Changes in the hormonal balance of postmenopausal women contribute to the accumulation of abdominal fat. This may result in hypertension, insulin resistance, lipid disorders and increased incidence of cardiovascular diseases [13]. An increase of only 1 cm in waist circumference is associated with a 2% increase in the risk of cardiovascular disease [14]. In several epidemiological studies, abdominal obesity is present in two-thirds of the women [14]. In Poland, the prevalence of metabolic syndrome (MS) after menopause, closely associated with obesity, is estimated at the level of 31–55% [15]. Regular physical activity is the best predictor for achieving long-term body weight control or reducing body weight and prevents cardiovascular events in menopausal women [15,16]. PA is a behavioral modality associated with physical self-esteem, body attractiveness and physical condition [17,18]. Weight loss is associated with a higher self-esteem in middle-aged women [19].

Self-esteem plays a fundamental role in people’s psychological well-being [20]. Self-esteem is based on many factors, such as social experience, lifestyle habits and family. It is culture-specific and cannot be assumed to eb universal. High self-esteem in women is related to a positive attitude towards life, better physical health, optimism and successful coping, while women with low self-esteem do not feel good about themselves and their social roles [21]. Higher self-esteem is negatively correlated with depressive symptoms [22,23].

The results of studies using the same tool to assess women’s self-esteem show that most women in Haiti over 45 years old had low self-esteem [24]. In Spain, about 35.6% of women reported low self-esteem [20], and in Switzerland, most menopausal women had moderate self-esteem [25]. Taken together, there are many unanswered questions regarding self-esteem in postmenopausal women. In postmenopausal women, self-esteem and health status are associated with feelings of body shame and lower psychological well-being compared with previous years [19]. Changes in body fat distribution and physiological changes in postmenopausal women could be associated with body dissatisfaction. Women often attribute their appearance with self-esteem; thus, this study was designed with two aims: the first aim was to assess self-esteem and physical activity levels in postmenopausal women. The second aim was to examine the association between physical activity levels and self-esteem in postmenopausal women. We hypothesized that high self-esteem is positively associated with high physical activity levels.

## 2. Materials and Methods

### 2.1. Participant

Women aged 45 and older from Silesia in Poland were invited to the survey study. Data was collected in women’s health clinics in Silesia in Poland between December 2021 and February 2022. The investigator met with patients in the clinics. Delivery and return of the study questionnaire took place in the clinics. The minimum sample size was estimated to be 97, and the total participant number in this study was 131, 18 were excluded.

Women were provided with information about the study. Participation was voluntary. Informed consent was obtained from all subjects involved, before the completion of a self-report questionnaire. The inclusion criteria were ages 45 and older and consent to participate in the study. The exclusion criteria were contraindications for physical activity, women who were still menstruating and have not passed one year since their last menstrual period and incomplete questionnaires. Participants were requested to fill out a questionnaire form containing sociodemographic data regarding smoking status, economic status, employment, hormonal therapy use, regular health screenings (breast ultrasound and cytology) and factors which would increase self-esteem. The next part of the questionnaire included the international Physical Activity Questionnaire (IPAQ-short form), the Rosenberg Self-Esteem Scale (RSES) and the Beck Depression Inventory (BDI). Ethics approval for this study was obtained from the Bioethical Committee of the Medical University of Silesia in Katowice (PCN/CBN/0022/KB/276/21).

### 2.2. Instruments

#### 2.2.1. General Data

Women were asked about age, educational level, parental status, smoking habits, place of living, hormonal therapy use, income, time since last menstruation and regular preventive gynecological examinations.

#### 2.2.2. International Physical Activity Questionnaire (IPAQ)—Short Form

IPAQ is a validated and reliable tool assessing physical activity level in adults aged 15–69 years. It consists of seven questions about the previous 7 days and asks about the frequency and duration of physical activity performed for at least 10 min. The IPAQ scoring protocol was used to evaluate weekly energy expenditure (MET-min/week) for walking, moderate and high physical activity and for total weekly energy expenditure. Vigorous PA was assigned to 8 MET, moderate PA to 4 MET and walking to 3 MET, and the result is expressed as MET-min/week. Women were divided into inactive (low PA level) and active (moderate and high AP levels). Women who did not meet the following criteria were inactive:−Three or more days of vigorous activity of >20 min per day;−Five or more days of moderate-intensity activity or walking of >30 min per day;−Five or more days of any combination of walking, moderate-intensity or vigorous-intensity activities achieving > 600 MET-min/week [26,27]. The Cronbach’s alpha was 0.83.

#### 2.2.3. Rosenberg Self-Esteem Scale (RSES)

RSES was used to evaluate self-esteem. This scale comprises ten statements, with four choices of answers on the Likert point scale (1 = strongly agree to 4 = strongly disagree). The scale has two factors. The first one comprises items that are positively worded (1, 2, 4, 6, 7) and the second factor (items: 3, 5, 8, 9, 10) that are negatively worded. Higher scores indicate higher levels of global self-esteem. The range of possible points is 10 to 40 points [28]. The Cronbach alpha was 0.81. In addition, the obtained results were converted into a sten scale, and the level of self-esteem was determined as low (sten 1–4), average (sten 5 and 6) or high (sten 7 to 10) [29].

#### 2.2.4. Beck Depression Inventory II (BDI–II)

BDI was used to determine the risk of depression. It is a 21-item self-reporting screening tool with a four-point Liker scale (0—not present, 3–severe symptoms). The total score ranged from 0 to 63 and was interpreted as 0 to 13 points—no depressive symptoms, 14 to 19—mild depression, 20–28—moderate depression and 29–63—severe depression [30]. The Cronbach α value was estimated at the level of 0.82.

### 2.3. Statistical Analysis

Statistical analysis was performed using the Statistica 10 (Statistica v10, StatSoft, Krakow, Poland). For measurable variables, arithmetic means, median and standard deviations were calculated. For qualitative variables, the percentage was calculated. To determine normality of data distribution, the Shapiro–Wilk test was used. The analysis of variance (ANOVA) was performed with post hoc Tuckey test. Pearson’s correlation coefficients were calculated to assess the association between total MET-min/week and total RSES value (self-esteem). The level of α = 0.05 was assumed as statistically significant.

## 3. Results

A total of 18 participants were excluded from the study due to refusal (*n* = 13) and incomplete questionnaires (*n* = 5). In total, 113 women were finally analyzed in the study. A mean age of participants was 58.81 ± 7.68. Characteristics of women are show in Table 1. A total of 15.04% of women had high educational level, and most participants had high school education (57.52%) and were married or living with a partner (71.68%). A total of 57.52% of postmenopausal women earned fewer than 4000 Polish zloty per month, 41.59% of participants were working, 10.62% used transdermal hormonal therapy and 19.47% were current smokers. Better health was a factor that could increase self-esteem, as reported by 56.64% of women. A total of 56.64% of participants reported that better health could increase their self-esteem. Higher income was a factor that could increase self-esteem in 26.55% of women.

The results show that 21.24% of women were physically inactive. A majority of women (78.76%) experienced no depressive symptoms. Most women presented average self-esteem (58.41%) (Table 2).

A detailed analysis of physical activity showed that postmenopausal women were mostly walking (M = 5.4 times per week). The average time per one walking session was 55 min.

A high PA level was performed on average 0.85 days per week and exercises lasted about 14.6 min per session (Table 3).

The total average score of self-esteem was significantly higher in women with insufficient PA levels (31.79) compared with physically active ones (29.74). The average scores of four RSES questions were shown to be significantly higher in inactive women (Table 4).

Most postmenopausal women, both physically inactive and active, had moderate self-esteem levels. Nobody from the insufficiently physically active group reported low self-esteem levels, however, self-esteem levels did not differ when comparing the difference between PA levels (Table 5).

Results show that the negative linear correlation was lower than 0.3 (r < −0.3) in walking, moderate and high-intensity PA and in the total IPAQ score. Therefore, a correlation between physical activity and self-esteem was not found (Table 6).

## 4. Discussion

Our study focused on physical activity and self-esteem in postmenopausal women.

In our study, women were involved in a high level of PA for approximately 15 min per session, averaging less than once a week. Moderate PA was performed more often than high PA, 1.4 times a week for at about 24 min. Overall, our results show that 21.24% of postmenopausal women were inactive. The results of the present study support the previous studies from Poland, where the percentage of sedentary postmenopausal women ranged from 21.85–27.03% [17,31]. Cabral PU et al. analyzed PA levels in Brazilian postmenopausal women and reported that 29.4% of participants were inactive. These results are also consistent with our study [32]. Those studies used the same and widely applied scale to assess PA levels, reducing the likelihood of systematic error. Many studies were conducted on postmenopausal women according to exercises and trainings, however, only few studies focused on the level of PA. Another study from Poland, conducted by Kroemeke A et al. on postmenopausal women, aimed to divide participants into active and inactive women according to daily steps. A total of 48.1% of participant did not meet the daily 10,000 steps as recommended, however, most of them performed walking every day and only 16.5% were inactive [33]. This higher percentage of active postmenopausal women can be caused by the fact that the sample was obtained from the University of the Third Age. It means that those participants had vast knowledge about physical activity and sport classes at the university. Such results could also suggest that PA education gives expected results and decreases the number of the inactive population. Barnet reported that difficulty reaching the exercise location was found to be an important factor influencing the PA of postmenopausal women [34,35]. Thus, walking is the alternative for those who cannot go to fitness clubs or other exercise locations. Results of our study show that walking was the most common type of PA chosen by postmenopausal women. Women were walking almost every day for an average of 55 min. Even 15–30 min per day of brisk walking is enough to be physically active at very modest levels. Most people are able to achieve it, and this in turn brings significant benefits for body composition, cardiometabolic health and aerobic fitness [36]. It is therefore important to find ways to encourage inactive postmenopausal women to walk regularly. On the other hand, walking has been shown to have little effect on preventing age-related bone loss in postmenopausal women because walking imparts low and insufficient loads to exceed the required threshold for skeletal adaptation [37]. However, other studies on postmenopausal women reported that walking combined with jogging or stair-climbing, or brisk walking at intensity around 75% of maximum oxygen uptake, can provide positive skeletal responses [38,39]. Kemmler W et al. reported that resistance exercises are the most powerful nonpharmaceutical fracture prevention strategy in postmenopausal women [40]. For this reason, despite the health benefits of walking, combined physical activity, including walking and resistance training to reduce bone loss and achieve skeletal benefits in postmenopausal women, should be recommended. Results published in Lancet report that by reducing physically inactive populations, life expectancy of the world’s population can be expected to increase by 0.68 years [41]. Other authors estimated that populations from the United States and Europe, aged 50 and older, could gain 1.3–3.7 years by becoming active [42]. Physical activity is the most effective nonpharmacological treatment for the improvement of general health [43,44]. Thus, all efforts should be made to promote physical activity, to increase the number of physically active people and decrease the inactive percentage of the population in the world.

In this study, postmenopausal women reported mostly moderate self-esteem (58.41%), and only 11.51% reported a low level of self-esteem. In our previous research, almost 32% of middle-aged women had low self-esteem [31], which was consistent with the study of Chedraui R et al., who assessed 149 Spanish women using the same tool, showing that 35.6% of postmenopausal women had low self-esteem [20]. In our study, the average value of self-esteem was 31.79 for inactive women and 29.74 in active women. This is consistent with the previous study of Sejourne N et al., who reported that postmenopausal women had self-esteems at the level of 32.58 [45]. However, our results are contrary to results from Switzerland, where the mean value of self-esteem in postmenopausal women was 38.86 [46]. All these studies evaluated self-esteem using the same tool, which made it easier to compare the data. However, our results show much lower self-esteem than in Switzerland. Moreover, contrary to our expectations, we failed to detect a positive correlation between physical activity level and better self-esteem. Our results show that inactive women reported higher self-esteem. Higher values of total MET-min/week in moderate and high-intensity PA were weakly correlated with worse self-esteem. Even though walking did not show a significant correlation, women who were walking more often also reported lower self-esteem.

No previous studies assessed self-esteem in postmenopausal women in regard to physical activity levels. There are mixed results about the relationship between physical activity and self-esteem. Recently, we have showed that middle-aged women with higher physical activity levels had better self-esteem [31]. In this study we concentrated on postmenopausal women, thus, it seems plausible to think that age could be a factor effecting self-esteem. However, Cherdaui P found no difference in self-esteem in relation to age in middle-aged Spanish women [20]. Ayers B et al. showed an indirect relation between PA and self-esteem and reported that self-esteem correlated with negative body image in women [47]. Elavsky et al. evaluated the effects of 4-month moderate-intensity walking and low-intensity yoga interventions on self-esteem in middle-aged women, and results did not show intervention effects relative to global self-esteem. These results also suggest that that self-esteem is less prone to change by PA interventions [18].

Our results are interesting and contrary to what we would have expected: lower physical activity is correlated with higher self-esteem in postmenopausal women. This may simply reflect the fact that postmenopausal women with low self-esteem decide to work out harder and be more physically active to increase their body image. However, they are perhaps not surprising given the global nature of this issue. They seem to point out that other factors are more important than physical activity in regard to self-esteem. It is also important to mention that according to Maslow’s pyramid, people with low self-esteem do not achieve their basic needs [48]. Self-esteem may be influenced by the social environment in which people live, their personal beliefs, individual goals or expectations. For this reason, such variables as social competence, experiences and psychological aspects are required to be considered in future studies [49,50].

### Limitations

Limitations of the study include a small sample size and a cross-sectional design. Despite these limitations, our results provide valuable information which can be used in future studies. More studies, incorporating other lifestyle and psychological aspects of self-esteem on postmenopausal women, are needed.

## 5. Conclusions

Physical activity levels among postmenopausal women in Poland are satisfactory, but further promotion of PA is needed. Most postmenopausal women present moderate self-esteem. However, it is multidimensional, and further studies are needed to explain the relationship between self-esteem and physical activity level in postmenopausal women.

## Figures and Tables

**Table 1 ijerph-19-09558-t001:** Characteristics of the study group (*n*-113).

Variable	*n*	%
Educational level		
elementary school	31	27.43
high school	65	57.52
university	17	15.04
Marital status		
single	6	5.31
married/with a partner	81	71.68
widowed	26	23.01
Place of living		
village	17	15.04
country	96	84.96
Economic status		
<40,000 Polish zloty	65	57.52
4000–6000 Polish zloty	44	38.94
>6000 Polish zloty	4	3.54
Employment		
working	47	41.59
not working	9	7.96
retired	38	33.63
retired and working	9	7.96
pension	10	8.85
Currently smoking		
yes	22	19.47
no	91	80.53
Hormonal therapy use		
yes	12	10.62
no	101	89.38
Regular screen (breast ultrasound, cytology)		
yes	83	73.45
no	30	26.55
What would increase your self-esteem?		
—higher educational level		
yes	8	7.08
no	105	92.92
—better health		
yes	64	56.64
no	49	43.36
—higher income		
yes	30	26.55
no	83	73.45
—changed place of living		
yes	11	9.73
no	102	90.27

**Table 2 ijerph-19-09558-t002:** Characteristics of the study group according to PA levels, depression symptoms and self-esteem.

Variable	*n*	%
IPAQ inactive	24	21,24
active	89	78.76
BDI		
no	89	78.76
mild	24	2124
moderate	0	0
severe	0	0
RSES		
low	13	11.51
average	66	58.41
high	34	30.08

IPAQ—International Physical Activity Questionnaire, BDI—Beck Depression Inventory and RSES—Rosenberg Self-Esteem Scale.

**Table 3 ijerph-19-09558-t003:** Characteristic of physical activity according to IPAQ.

Variable	M	SD	Min	Max
Time of PA (min)				
high PA level	14.60	23.07	0.00	90.00
moderate PA level	24.34	30.50	0.00	120.00
walking	55.00	25.72	30.00	120.00
Day of PA				
high PA level	0.85	1.38	0.00	6.00
moderate PA level	1.39	1.76	0.00	7.00
walking	5.40	1.40	1.00	7.00
MET-min/week high PA	267.61	463.65	0.00	2160.00
MET-min/week moderate PA	243.36	315.89	0.00	1440.00
MET-min/week walking	1032.49	662.07	148.50	2772.00
MET-min/week total	1543.46	1060.92	198.00	5319.00

M—mean value, SD—standard deviation, Min—minimum value, Max—maximum value, PA—physical activity and IPAQ—International Physical Activity Questionnaire.

**Table 4 ijerph-19-09558-t004:** Characteristics of self-esteem according to RSES according to physical activity level.

Self-Esteem	Insufficient PA Level (*n* = 24)				Sufficient PA Level (*n* = 89)				
	M	SD	Min	Max	M	SD	Min	Max	*p*
RSES1	3.50	0.51	3	4	3.24	0.43	3	4	0.012
RSES2	3.50	0.51	3	4	3.06	0.23	3	4	0.000
RSES3	2.88	0.34	2	3	2.67	0.52	1	3	0.077
RSES4	3.04	0.20	3	4	2.92	0.66	1	4	0.552
RSES5	2.88	0.34	2	3	2.89	0.41	1	3	0.526
RSES6	3.50	0.51	3	4	3.16	0.56	2	4	0.009
RSES7	3.50	0.51	3	4	3.24	0.43	3	4	0.012
RSES8	2.58	0.58	2	4	2.46	0.54	2	4	0.349
RSES9	3.04	0.20	3	4	3.07	0.54	2	4	0.763
RSES10	3.38	049	3	4	304	0.66	2	4	0.028
RSES Total	31.79	2.93	29	38	29.74	2.98	25	38	0.003

M—mean value, SD—standard deviation, Min—minimum value, Max—maximum value, RSES 1–10, Rosenberg Self-esteem Scale questions 1–10 and PA—physical activity.

**Table 5 ijerph-19-09558-t005:** Self-esteem level according to PA levels.

Self-Esteem	Insufficient PA Level		Sufficient PA Level		*p*
	*n*	%	*n*	%	0.201
low	0	0.00%	13	14.61%	
moderate	16	66.67%	51	57.30%	
high	4	33.33%	25	28.09%	

PA—physical activity.

**Table 6 ijerph-19-09558-t006:** Correlation between physical activity intensity and self-esteem.

Variable	Self-Esteem	
MET-min/week:	r	*p*
walking	−0.074	0.439
moderate- intensity PA	−0.282	0.002
high-intensity PA	−0.256	0.006
total IPAQ score	−0.241	0.01

PA—physical activity and IPAQ—International Physical Activity Questionnaire.

## Data Availability

Not applicable.

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
