# Peer review of "The Relationship between Postmenopausal Women’s Self-Esteem and Physical Activity Level—A Survey Study from Poland"

_ijerph, 2022, doi:10.3390/ijerph19159558_

Round 1

Reviewer 1 Report

Thank you for the opportunity to review this manuscript. Overall, I feel the English language and style need addressed throughout. Also, the study setting has not been made clear and further detail is required to improve reporting of the results. As such, revisions are required. 

1)    Title and Abstract: suggest study design to be reported.

2)    Introduction:  Please consider revising the wording and paragraph structure to strengthen the readability of your introduction.  For example, line 32 reads: “WHO recommends for adults at least 150 minutes of moderate physical activity per week”. Instead consider: The WHO recommend that adults achieve a minimum of 150 minutes of moderate physical activity per week. Also, the first time you write WHO in the introduction, please can you write this in full? For example, World Health Organization (WHO). The WHO reference is incorrectly displayed in the reference section.  Please correct. Further, please mention the risk of depression for this population to help the reader understand why it was necessary to use the BDI as a measure.

3)    Methods section: study setting - The sentence on line 75 does not make sense. For example, “On arrival to women’s health clinics in Silesia and via internet.” Please clarify what you mean and if this sentence is related to participant recruitment, explain how participants were recruited via the internet. Also, please provide detail regarding the study recruitment period and further information regarding the data collection processes (who was involved etc.). In section 2.2.1 IPAQ – short form, please report how the physical activity cut off points of Inactive and Active were categorised from the MET-min/week from the IPAQ.

4)    Results: The reference to HT in the participant description and characteristics table needs to be made clear. Please correct the characteristics table to show only one heading for marital status. To strengthen your results, please consider reporting the data distribution test results.  

5)    Discussion: Consider starting the discussion session with your main findings as opposed to the existing literature. Please revise the wording and English language throughout to strengthen the readability of your discussion section.

Author Response

Reviewer 1:

Thank you for the opportunity to review this manuscript. Overall, I feel the English language and style need addressed throughout. Also, the study setting has not been made clear and further detail is required to improve reporting of the results. As such, revisions are required. 

1)  Title and Abstract: suggest study design to be reported.

Response:

We thank the reviewer for time necessary to review the manuscript and for all very important suggestions and comments. We thank for this comment. We have changed title into “Self-Esteem and Physical Activity in Postmenopausal Women – a survey study from Poland”, and added details to abstract.

2)    Introduction:  Please consider revising the wording and paragraph structure to strengthen the readability of your introduction.  For example, line 32 reads: “WHO recommends for adults at least 150 minutes of moderate physical activity per week”. Instead consider: The WHO recommend that adults achieve a minimum of 150 minutes of moderate physical activity per week. Also, the first time you write WHO in the introduction, please can you write this in full? For example, World Health Organization (WHO). The WHO reference is incorrectly displayed in the reference section.  Please correct. Further, please mention the risk of depression for this population to help the reader understand why it was necessary to use the BDI as a measure.

Response:

We thank the reviewer for this important suggestion. We have revised the sentence as recommended and wrote: “The WHO recommend that adults achieve a minimum of 150 minutes of moderate physical activity per week”.

-We have explained the shortcut WHO, and changed WHO reference into: https://www.who.int/news-room/fact-sheets/detail/physical-activity

- We have also added to Introduction that “Higher self-esteem is negatively correlated with depressive symptoms”, as recommended. We are grateful for this important comment.

3)    Methods section: study setting - The sentence on line 75 does not make sense. For example, “On arrival to women’s health clinics in Silesia and via internet.” Please clarify what you mean and if this sentence is related to participant recruitment, explain how participants were recruited via the internet. Also, please provide detail regarding the study recruitment period and further information regarding the data collection processes (who was involved etc.). In section 2.2.1 IPAQ – short form, please report how the physical activity cut off points of Inactive and Active were categorised from the MET-min/week from the IPAQ.

Response:

We very much appreciate this helpful comment and agree that the reader could be confused reading information about participants recruitment. We have revised this part as suggested. In the Internet there was only an invitation to participate in the study while being in the women’s health clinic, however this is maybe not important because the recruitment process was carried out only in clinics. We have added such information: “Women aged 45 and older from Silesia in Poland were invited to the survey study. Data was collected in women’s health clinics in Silesia in Poland between December 2021 and February 2022. The investigator met with patients in the clinics. Delivery and return of the study questionnaire took place in the clinics.”.

                  We thank the reviewer for the comment about IPAQ. We have added such information:

“Women who did not meet the following criteria were inactive:

-three or more days of vigorous activity of > 20 min per day OR

-five or more days of moderate-intensity activity or walking of > 30 min per day or

-five or more days of any combination of walking, moderate-intensity or vigorous intensity activities achieving > 600 MET-min/week

We would like to explain how the PA groups were divided, however we decided not to write it in the manuscript, but to add references.

According to IPAQ scoring protocol, physical activity level is divided into low, moderate and high:

High PA level:

Vigorous-intensity activity on at least three days and accumulating at least 1500 MET-minutes/week, or

Seven or more days of any combination of walking, moderate-intensity or vigorous intensity activities

achieving a minimum of at least 3000 MET-minutes/week

Moderate PA level:

Any one of the following three criteria:

Three or more days of vigorous activity of at least 20 min per day OR

Five or more days of moderate-intensity activity or walking of at least 30 min per day or

Five or more days of any combination of walking, moderate-intensity or vigorous intensity activities

achieving a minimum of at least 600 MET-min/week.

Low PA level:

This is the lowest level of physical activity. Those individuals who not meet criteria for categories 2 or 3 are considered low/inactive.

4)    Results: The reference to HT in the participant description and characteristics table needs to be made clear. Please correct the characteristics table to show only one heading for marital status. To strengthen your results, please consider reporting the data distribution test results.  

Response:

We thank the reviewer for noticing this mistake. We have changed one heading into economic status. We have added detailed to material and methods information about questions in sociodemographic survey, revised results and added information about HT – it was transdermal hormonal therapy.  

5)    Discussion: Consider starting the discussion session with your main findings as opposed to the existing literature. Please revise the wording and English language throughout to strengthen the readability of your discussion section.

      Response: Thank you for this suggestion, we have started the discussion from our results as recommended. A native English speaker has proofread the manuscript.

Author Response

Reviewer 2

The manuscript entitled "Self-Esteem in Inactive Postmenopausal Women ‘’ is interesting, but the

manuscript needs to be improved.

I have the following comments to improve the manuscript:

Introduction

- the sample size is small, so it is worth writing what this problem looks like in the world

Response

We thank the reviewer for all suggestions and appreciate the encouraging comments.

We have added information about the self-esteem level in different countries. Physical inactivity is presented as a world global problem.

Material and Methods

- please explain how the data was collected over the internet our through some forums for women

or maybe in some other way. Please expand this section on data collection

- please provide specific inclusion and exclusion criteria in the participant section

Response:

We thank the reviewer for this suggestion. We did not use any google form to collect data but to invite women to participate in the study while visiting health clinic. We agree that this sentence would be confusing so we decided to delete it from the manuscript. 

We have revised this part into:

Women aged 45 and older from Silesia in Poland were invited to the survey study. Data was collected in women’s health clinics in Silesia in Poland between December 2021 and February 2022. The investigator met with patients in the clinics. Delivery and return of the study questionnaire took place in the clinics.

            As recommended, we have added inclusion and exclusion criteria:

The inclusion criteria were age 45 and older, and consent to participate in the study. The exclusion criteria were contraindication for physical activity, women who were still menstruating and have not passed one year since their last menstrual period, and incomplete questionnaires

Reviewer 3 Report

This is an interesting study about of physical activity and self esteem. However, the explanation of the research method was insufficient, and the interpretation of the results was wrong.

The major revision is required.

1. Extensive english proofreading is needed.

2. abstract:

-Total participants should be included. It must be presented as a survey study in title and abstract.

- There was a negative correlation between total IPAQ score and self-esteem (r=-0.241, p=0.01). ->The r value below 0.3 is consider low correlation. Should be revised the results.

3. Introduction.

Definition of physical inactivity is need .

In phrases that use percentages, be clear about the percentages’s meaning. Please clarify the definition of inactivity and the percentage

ex) what does 70% mean? 

-In some countries, the level of inactive adult population is 70% [3]

Physical inactivity is responsible for approximately 10% of type 2 diabetes, coronary heart disease, an..

3. Mathods.

- Sample size determination should be included in this survey study.

-In table, add specific explanation. pln? HT? SES1..10

ex) Material statu /<40000 pln/HT use

Table 4?Add abbreviation for table 4 and specific explanation.

Table 6. r below 0.3 is consider low correlation. This is not important data because it is meaningless. Therefore figure 2 is not need in this research.

The meaning of the data and the conclusion of this paper are different.

Author Response

Reviewer 3 :

This is an interesting study about of physical activity and self esteem. However, the explanation of the research method was insufficient, and the interpretation of the results was wrong.

The major revision is required.

1.Extensive english proofreading is needed. –

Response:

We thank the reviewer for all suggestions and appreciate the encouraging comments.

We do apologize for English mistakes. A native English speaker has proofread the manuscript.

  1. abstract:

-Total participants should be included. It must be presented as a survey study in title and abstract.

Response:

We thank the reviewer for this suggestion. As recommended, we have added to title information that it was a survey study and in abstract we have wrote:

“A survey research was conducted on postmenopausal women aged M=58,81 ± 7,68 in women’s health clinics in Silesia in Poland. Total participants number was 131, 18 were excluded.”

- There was a negative correlation between total IPAQ score and self-esteem (r=-0.241, p=0.01). ->The r value below 0.3 is consider low correlation. Should be revised the results.

Response:

Thank you for this important comment. As recommended, we have changed “low correlation” into “a lack of correlation”.

  1. Introduction.

Definition of physical inactivity is need . 

In phrases that use percentages, be clear about the percentages’s meaning. Please clarify the definition of inactivity and the percentage

  1. ex) what does 70% mean? 

-In some countries, the level of inactive adult population is 70% [3]

Physical inactivity is responsible for approximately 10% of type 2 diabetes, coronary heart disease,

Response:

We thank the reviewer for this comment. We have added definition of physical activity and inactivity. We agree that this is a very important information for the reader. We have written:

Physical activity (PA) is defined as a behavior that involves human movement, resulting in physiological attributes including increased energy expenditure and im-proved physical fitness [Daley A., MacArthur C., McManus R., Stokes-Lampard H., Wilson S., Roalfe A., Mutrie N. Factors associated with the use of complementary med-icine and non-pharmacological interventions in symptomatic menopausal women. Cli-macteric. 2006;9:336–346. doi: 10.1080/13697130600864074]. Adults are classified as inac-tive if they did not perform any session of light to vigorous leisure time PA of at least 10 minutes per day [Pleis JR, Lucas JW, Ward BW. Summary health statistics for U.S. adults: National Health Interview Survey, 2008. National Center for Health Statistics. Vital Health Stat. 2009;10(242).} Another definition says that individuals who walk less than ap-proximately 3000 steps or expend less than 1.5 kcal/kg/day are considered to be inactive [Canizares M, Badley EM. Generational differences in patterns of physical activities over time in the Canadian population: an age-period-cohort analysis. BMC Public Health. 2018;18(1):304.

According to percentages:

what does 70% mean? -In some countries, the level of inactive adult population is 70% [3]).

WHO has reported that as countries developed economically, levels of inactivity increase and is as high as 70%, due to changing patterns of transportation, increased use of technology and urbanization. This is surprisingly, but it means that 70% of adult population in some developed countries does not meet WHO physical activity recommendations and they are inactive.

We have revised the sentence into:

“In some countries, due to changing patterns of transportation, increased use of technology and urbanization, the level of inactive adult population increased to 70%”

Physical inactivity is responsible for approximately 10% of type 2 diabetes, coronary heart disease,

We have added interpretation of this sentence: “Elimination of physical inactivity would remove approximately 10% of major CHD, type 2 diabetes, and breast cancer”.

  1. Mathods.

- Sample size determination should be included in this survey study.

-In table, add specific explanation. pln? HT? SES1..10

  1. ex) Material statu /<40000 pln/HT use

Table 4?Add abbreviation for table 4 and specific explanation.

Response:

We agree with the reviewer. The minimum sample size was estimated to be 97 – we have added this information to material and method section.

According to your suggestion in all tables, abbreviations were explained under each table.

Table 6. r below 0.3 is consider low correlation. This is not important data because it is meaningless. Therefore figure 2 is not need in this research.

Response:

We are grateful for this comment as it points to an important rationale of this study. As recommended, we have deleted figures with correlations with r < 0.3. Also, in results we have changed description of the results.

The meaning of the data and the conclusion of this paper are different.

Response:

We appreciate this comment and the opportunity to revise the conclusions.

We have changed the conclusions: “Physical activity levels among postmenopausal women in Poland are satisfactory, but further promotion of PA is needed. Most postmenopausal women present moderate self-esteem. However, it is multidimensional, and further studies are needed to explain the relationship between self-esteem and physical activity level in postmenopausal women”.

We thank the reviewer one more time for such important comment.
